# The Responses to Long-Term Water Addition of Soil Bacterial, Archaeal, and Fungal Communities in a Desert Ecosystem

**DOI:** 10.3390/microorganisms9050981

**Published:** 2021-04-30

**Authors:** Ying Gao, Xiaotian Xu, Junjun Ding, Fang Bao, Yashika G. De Costa, Weiqin Zhuang, Bo Wu

**Affiliations:** 1Institute of Desertification Studies, Chinese Academy of Forestry, Beijing 100091, China; xuxiaotian@caf.ac.cn (X.X.); baofang@caf.ac.cn (F.B.); 2Beijing Academy of Forestry and Pomology Sciences, Beijing 100093, China; 3Key Laboratory of Dryland Agriculture, Ministry of Agriculture, Institute of Environment and Sustainable Development in Agriculture, Chinese Academy of Agricultural Sciences, Beijing 100081, China; dingjunjun@caas.cn; 4Department of Civil and Environmental Engineering, The University of Auckland, Auckland 1023, New Zealand; ydec845@aucklanduni.ac.nz (Y.G.D.C.); wq.zhuang@auckland.ac.nz (W.Z.)

**Keywords:** global climate change, water addition, Illumina sequencing, desert, microbial community

## Abstract

The response of microbial communities to continual and prolonged water exposure provides useful insight when facing global climate changes that cause increased and uneven precipitation and extreme rainfall events. In this study, we investigated an in situ manipulative experiment with four levels of water exposure (ambient precipitation +0%, +25%, +50%, and +100% of local annual mean precipitation) in a desert ecosystem of China. After 9 years of water addition, Illumina sequencing was used to analyze taxonomic compositions of the soil bacterial, archaeal, and fungal communities. The results showed significant increases in microbial biomass carbon (MBC) at higher amended precipitation levels, with the highest values reported at 100% precipitation. Furthermore, an increase in the bacterial species richness was observed along the water addition gradient. In addition, the relative abundance of several bacterial phyla, such as Proteobacteria, significantly increased, whereas that of some drought-tolerant taxa, including Actinobacteria, Firmicutes, and Bacteroidetes, decreased. In addition, the phyla Planctomycetes and Nitrospirae, associated with nitrification, positively responded to increased precipitation. Archaeal diversity significantly reduced under 100% treatment, with changes in the relative abundance of Thaumarchaeota and Euryarchaeota being the main contributors to shifts in the archaeal community. The fungal community composition was stable in response to water addition. Results from the Mantel test and structural equation models suggested that bacterial and archaeal communities reacted contrastingly to water addition. Bacterial community composition was directly affected by changing soil moisture and temperature, while archaeal community composition was indirectly affected by changing nitrogen availability. These findings highlight the importance of soil moisture and nitrogen in driving microbial responses to long-term precipitation changes in the desert ecosystem.

## 1. Introduction

Anthropogenic emission of greenhouse gases increase the Earth’s surface temperature and significantly affects the global hydrological cycle [1,2]. According to global circulation models (GCMs), precipitation and frequency of extreme rainfall events will increase during the 21st century [3]. Such altered precipitation patterns are likely to significantly impact the soil microbial activity and plant productivity in water-limited regions (e.g., arid and semi-arid regions), as water availability is the main limiting factor [4,5]. Deserts, as typical ecosystems in arid regions, cover approximately 19% of the Earth’s terrestrial surface [6]. They support ca. 6% of the human population and store almost 15% of the Earth’s organic carbon [7,8]. Considering their relevance to important ecosystem services, it is essential to understand the consequential effect of precipitation changes on desert ecosystems.

As conditions of desert surface soil are extremely harsh (e.g., oligotrophy, water scarcity, large temperature fluctuations, and high ultraviolet radiation [9]), plant and animal life is limited, with soil microbes being the most productive components in the ecosystem [10]. However, desert microbial communities are highly vulnerable to precipitation pattern changes, as substrate availability in desert soil is predominately controlled by soil’s water availability. Thus, increasing precipitation induced by global climate change may have profound impacts on the structures of indigenous microbial communities [5,11]. Since microbes are dominant drivers of biogeochemical cycling in ecosystems [5,11,12], changes in microbial community composition might ultimately cause changes in ecosystem functions.

Based on previous reports, increased precipitation generally enhances microbial biomass and richness in desert soils [4,13,14]. Studies on the response of microbial community composition to precipitation increase have found that bacteria and fungi performed differently. Specifically, bacteria generally responded faster than fungi to water availability changes [15,16]. This was attributed to differences in their cell structure and physiological traits [17,18]. Fungal hyphae can tolerate low water availability due to their ability to access water from distant micropores with their extensive hyphae [19,20,21]. However, compared with bacteria and fungi, archaea in desert environments have, so far, received considerably less attention. Similar to soil bacteria and fungi, archaea are major components of belowground diversity and participate in various biogeochemical processes, such as organic carbon degradation [22] and methane production [23]. Recent studies have suggested that archaea are potentially crucial for nitrogen cycling in desert soils [24,25]. Additionally, despite decades of research on soil microbial community in desert ecosystems, most studies have only made use of traditional technologies, such as denaturing gradient gel electrophoresis (DGGE) analysis [26] and fatty acid methyl ester (FAME) analysis [17]. More precise methods in this field (i.e., high-throughput sequencing) are limited, particularly in descriptions of the microbial community composition under precipitation changes [20,27]. Compared with traditional technologies, high-throughput sequencing can provide extensive data on microbial community diversity and dynamics. Thus, enabling a more in depth resolution of changes in the microbial community, to deepen our knowledge of desert ecosystems and their response to climate change.

As microbial responses are complex in their direction and magnitude, they tend to be inconsistent over time [28]. For example, one study conducted in the Chihuahuan Desert found that in the first two years, responses of the microbial community due to increased precipitation levels were insignificant, but after three to five years; higher microbial biomass, arbuscular mycorrhizae abundance, and soil enzyme activities were observed [17], suggesting a prolonged response by microbial communities. This is most likely caused by microbial resistance to environmental disturbances [4]. Furthermore, short-term precipitation pulses may induce increased microbial decomposition rates; however, it would be attenuated by long-term environmental adaptation of microbial communities [17]. Since the effects of precipitation increase are evident in the long-term, experiments over longer timescales are needed to examine the effects of such changes on soil microbes in desert ecosystems.

In this context, we experimented with four levels of water addition (ambient precipitation +0%, +25%, +50%, and +100% of local annual mean precipitation) in a desert in China, to examine the effects of long-term precipitation increases on the soil microbial community. The experiment started in May 2008, and included the determination of various soil physico-chemical parameters to detect changes in environmental variables. We adopted Illumina sequencing of 16S rRNA gene amplicons to individually distinguish the taxonomic compositions of bacterial, archaeal, and fungal communities, aiming to answer the following three questions: (1) How do bacterial, archaeal, and fungal communities respond to long-term water addition in a desert ecosystem? (2) What are the specific taxa that are responsible for the water addition-induced alterations in these communities? (3) What are the key environmental factors controlling variations of these communities?

## 2. Materials and Methods

### 2.1. Study Site

The study was conducted in a desert ecosystem on the northeast edge of the Ulan Buh Desert (40°24’ N, 106°43’ E, 1050 m a. s. l.). This site is located in Dengkou County, Inner Mongolia, China, and is managed by the Experimental Center of Desert Forestry of the Chinese Academy of Forestry. This area is characterized by an arid continental climate, with a mean annual temperature of 7.8 °C and mean annual precipitation of 145 mm (1961–2000); 77.5% of the annual precipitation occurs between June and September. The mean annual potential evaporation is 2327 mm [29] and precipitation during the growing season is expected to increase in the future [30]. The vegetation type is temperate desert shrubland with a community cover of 20–30%. The plant community develops on nabkhas (a sand dune that forms around vegetation) and is solely dominated by the widely distributed desert shrub species, *Nitraria tangutorum,* while the semi-shrub *Artemisia ordosica*, the perennial grass *Psammochloa villosa*, the annual grasses *Agriophyllum squarrosum* and *Corispermum mongolicum* can also be found. Many *N. tangutorum* nabkhas are scattered on red clay ground to form a mosaic desert landscape. The soils are classified as sandy soil and gray-brown desert soil (Cambic Arenosols and Luvic Gypsisols in FAO taxonomy).

### 2.2. Experimental Design

In 2008, 16 plots with a diameter of 12 m were established within the experimental area in a randomized block design. Plots were separated by a buffer zone of more than 5 m to avoid interactions. Within each plot, one natural nabkha was situated in the center. The height of the nabkhas ranged within 1.18–1.40 m, with base-diameters of 5.75–8.83 m. We selected nabkhas with similar growing conditions and applied four water addition levels: control (C) = ambient precipitation, W25 = ambient precipitation +25% of local annual mean precipitation, W50 = ambient precipitation +50% of local annual mean precipitation, and W100 = ambient precipitation +100% of local annual mean precipitation. The treatments were equally applied on the 15th of each month from May to September, with four replicates per treatment. The water used in the experiments were pumped from a nearby well into a tank and then sprayed onto the plots using an irrigation system installed on the top of each nabkha. Since the two spraying arms of the irrigation system can rotate freely, water could be uniformly sprinkled over the treatment area (Appendix A). To reduce water evaporation, water was applied to the plots in the morning, when the air temperature was relatively low. The groundwater level was below 5 m and therefore did not affect plant growth. The experimental design was similar to another experiment conducted in Gansu Province, where more detailed information can be found in Song et al., 2012 [31].

Soil samples were collected on 22 September 2016, when soil moisture had stabilized according to the soil moisture records (Appendix A). Since our objective is to estimate average changes of microbial community over the entire nabkha, to minimize effects of inequal root distribution, sampling locations were selected with similar environmental conditions in all nabkha. In addition, in each plot, five representative points (four in the corners and one in the center) of the topsoil (0–20 cm in depth) were randomly collected by a 5 cm drill. Prior to each sampling event, the drill was cleaned, washed with sterile water, and air dried. Samples from the same plot were pooled to obtain one composite sample, packed in polyethylene bags, immediately stored in a cooler with ice packs, and transferred to the laboratory. The composite samples were sieved through a 2 mm mesh sieve, and visible roots, residues, and stones were removed. Subsequently, the samples were divided into three parts. One part was stored at 4 °C for the measurement of soil dissolvable inorganic nitrogen, including nitrate (NO_3_^−^) and ammonium (NH_4_^+^), soil microbial biomass C (MBC), and N (MBN). The second part was air-dried to determine soil pH, total carbon (TC), total nitrogen (TN), and total phosphorous (TP), while the third part was stored at −80 °C for soil DNA extraction.

### 2.3. Measurement of Biotic and Abiotic Factors

A permanent quadrat of 1 × 1 m was established in each plot to conduct plant surveys, recording plant species and species richness. The percentage of plant cover was measured in each quadrat using a 1 × 1 m metal pane with 100 equal grids (10 × 1.0 cm) by counting the grid junctions whose vertical projections overlapped with the plant species. Plant coverage was visually estimated for species not present at the junctions or present at the junctions but occupying small areas in the quadrat.

The long-term soil temperature and soil moisture dynamics at a depth of 20 cm were automatically monitored hourly by EM50 data logger systems (Decagon, WA, USA) in four typical plots within each treatment; the values were averaged over the entire month. Soil pH was determined in 1 mol L^−1^ KCl extracts. Extractable ammonium (NH_4_^+^) concentrations and extractable nitrate (NO_3_^−^) concentrations were determined by extracting 10 g of fresh soil samples with 50 mL 2 mol L^−1^ KCl. Subsequently, the KCl extracts were analyzed with a continuous flow analyzer (Futura, Alliance Instruments, Paris, France). Soil organic carbon contents were determined using a sulfuric acid and aqueous potassium dichromate (K_2_Cr_2_O_7_) mixture with external heating [32]. Soil temperature, soil moisture, NH_4_^+^ concentrations, NO_3_^−^ concentrations, and soil organic carbon were measured in 2016 (year 9 of the experiment). Soil microbial biomass C (MBC) and N (MBN) levels were determined in 2017 (year 10 of the experiment) by using the chloroform fumigation-extraction method [33].

### 2.4. Soil DNA Extraction and Sequencing

Total DNA was extracted from 0.5 g of soil per replicate, using a Power Soil DNA kit (Mo Bio Laboratories, Carlsbad, California, USA) according to the manufacturer’s instructions. All samples had a 260/280 ratio between 1.6 and 2.0. Final DNA concentrations were quantified using the Quant-IT Pico Green dsDNA Kit (Invitrogen Molecular Probes Inc., Oregon, USA). The V4 hypervariable region of bacterial 16S rRNA gene was amplified with the barcoded primer set consisting of 515F (5′-GTGCCAGCMGCCGCGGTAA-3′) and 806R (5′-GGACTACHVGGGTWTCTAAT-3′). The V4 hypervariable region of archaea 16S rRNA gene was amplified using the barcoded primer set of Arch519F (5′-CAGCCGCCGCGGTAA-3′) and Arch915R (5′-GTGCTCCCCCGCCAATTCCT-3′). The ITS1 region of the fungal rRNA gene was amplified by the primers ITS5-1737F (5′-GGAAGTAAAAGTCGTAACAAGG-3′) and ITS2-2043R (5′-GCTGCGTTCTTCATCGATGC-3′). Bacterial, archaeal, and fungal amplicons were sequenced on an Illumina HiSeq2500 platform [34] at Novogene Bioinformatics Technology Co. Ltd. (Beijing, China). All Sequencing libraries were generated using the TruSeq^®^ DNA PCR-Free Sample Preparation Kit (Illumina, San Diego, CA, USA), following the manufacturer’s recommendations and index codes were added. The library quality was assessed on the Qubit^®^ 2.0 Fluorometer (Life Technologies, Carlsbad, CA, USA) and the Agilent Bioanalyzer 2100 system (Agilent Technologies Inc., Santa Clara, CA, USA). Finally, the library was sequenced on an Illumina HiSeq2500 platform, and 250 bp paired-end reads were generated. Paired-end reads were assigned to samples based on their unique barcodes and truncated by cutting off the barcode and primer sequence. Paired-end reads were merged using FLASH (V1.2.7, http://ccb.jhu.edu/software/FLASH/, accessed on 1 October 2017) and merged sequences were quality filtered according to the QIIME (V1.7.0, http://qiime.org/index.html, accessed on 31 October 2017) [35] with default settings. The tags were compared with the reference database (Gold database, http://drive5.com/uchime/uchime_download.html, accessed on 10 November 2017) using UCHIME algorithm (UCHIME Algorithm, http://www.drive5.com/usearch/manual/uchime_algo.html, accessed on 25 October 2017) to detect chimera sequences, and subsequently, the chimera sequences were removed. Sequence analyses were performed by UPARSE (v7.0.1001, http://drive5.com/uparse/, accessed on 1 December 2017 [36]). Sequences with ≥97% similarity were assigned to the same OTUs. For each OTU, the most abundant sequence was regarded as its representative sequence, and subsequently, representative sequences were assigned taxonomy information with the Silva database as a reference. Non-bacterial sequences were discarded. The final number of bacterial sequence reads ranged from 57,913 to 74,977 per sample. To correct the bias caused by different sequencing depths, the sequence data were normalized to 57,913 sequences per sample. For archaeal samples, non-archaeal sequences were discarded after the step of assigning representative sequence taxonomy information with the Silva database [37]. Finally, the number of archaeal sequence reads ranged from 24,965 to 53,260 per sample. Archaeal communities were rarefied to 24,965 sequences. For fungi samples, representative sequences were assigned taxonomy information with the UNITE database as a reference. The number of fungal sequence reads ranged from 62,444 to 82,375 per sample; fungal communities were rarefied to 62,444 sequences. In total, 40 bacterial, 7 archaeal, and 6 fungal phyla were detected. Sequence reads generated in this study were archived in the sequence read archive database of the National Center for Biotechnology Information under accession number PRJNA563974.

### 2.5. Statistical Analysis

Microbial alpha- diversity was assessed based on the observed species (species richness), the Shannon–Wiener index (H), and Pielou’s evenness (J). Differences in microbial community composition (Bray–Curtis dissimilarity) among the different treatments were assessed by permutational multi-variate ANOVA (PERMANOVA) and visualized using principal coordinate analysis (PCoA). One-way ANOVA was conducted to compare the effects of different water addition treatments. Post hoc analyses were performed using Fisher’s least significant difference (LSD) test and the Mantel test was used to test relationships between plant/soil variables and the microbial community compositions. Structural equation modeling (SEM) was performed to specifically test the direct and indirect effects of water addition on the compositions of soil microbial communities (as assessed by PCo1 of the Bray–Curtis dissimilarity matrix). Before conducting SEM analysis, we hypothesized that water addition would impact the bacterial/archaeal/fungi communities, either directly by increasing water availability, or indirectly through changing plant coverage and soil variables (e.g., TC, TN, and NO_3_^−^), based on our ANOVA and Mantel test results. The data matrix was fitted to the model using the maximum likelihood estimation method [38]. The overall goodness-of-fit of our model was characterized by a non-significant chi-square test (*p* > 0.05), a low Akaike information criteria (AIC), a low root-mean-square error of approximation (RMSEA < 0.05 and *p* > 0.1), and a comparative fit index (CFI) > 0.95. The Benjamini–Hochberg method (false discovery rate; FDR) was applied to correct the *P*-values after performing multiple comparisons [39].

All statistical analyses and figures described above were completed with R 3.3.1 (www.R-project.org, accessed on 10 December 2020) [40]. The vegan package [41] was used for computing the Bray–Curtis dissimilarity, the Mantel test, and the PCoA. The package pairwise.adonis was used for conducting the PERMANOVA [42]. The lavaan package [43] was employed for SEM, and the ggplot2 package [44] for generating the figures.

## 3. Results

### 3.1. Soil and Plant Variables in Different Water Addition Treatments

As expected, soil moisture increased from 3.2% (control) to 3.8% (W50) and 6.8% (W100) with increased water addition (Table 1), with increment rates of 1, 21.5, and 114.0%, respectively. However, our study site showed low total carbon and nitrogen values which were significantly enhanced by 100% water addition compared to the control and other water addition treatments (Table 1). Average increases in total carbon (0.06 g TC/kg soil) and total nitrogen (0.04 g TN/kg soil) corresponded to 60.0% of total soil carbon and 28.6% of total nitrogen. Although plant coverage did not demonstrate significant changes, it increased along the water addition gradient (from 25.0% in control plots to 33.8% in W100 plots, *p* = 0.20). In addition, total soil phosphorus, pH, nitrate concentration (NO_3_^−^), and plant species did not show any significant differences among different treatments. Ammonium concentrations (NH_4_^+^) were below the detection limit in all treatments.

### 3.2. Soil Microbial Biomass, Diversity, and Community Composition

MBC increased along the water addition gradient and reached maximum values in the W100 plots (from 11.7 ± 4.1% in control to 73.7 ± 6.9% in W100 plots), being 6.29 times higher than in the control (Appendix A). However, MBN could not be detected because of the low content.

To further dissect the changes in soil bacterial, archaeal, and fungal communities after water addition, we evaluated the alpha diversities in these communities. For bacteria, water addition treatments significantly increased species richness by 17.5% in W100 plots (Figure 1a). Although the Shannon diversity index and Pielou’s evenness index increased compared to the control, these differences were insignificant (Figure 1b,c). For archaea, water addition did not significantly affect species richness (Figure 1d), but significantly decreased the Shannon diversity index by 7.3% and Pielou’s evenness index by 8.4% in W100 plots (Figure 1e,f). For fungi, water addition had no significant impact on species richness, Shannon diversity index, and Pielou’s evenness index compared with the control plots (Figure 1g–i).

Unconstrained principal coordinate analyses (PCoAs) based on the Bray–Curtis distance matrix of OTU relative abundances were performed to investigate patterns of microbial communities (Figure 2a–c). The PCoA plots showed that at the OTU_0.03_ level, both bacterial and archaeal communities shifted in composition by water addition treatments whereas fungal communities in W100 plots tended separate from the control, though with overlap. Bacterial axes 1 and 2 explained 29.4% and 12.5% of the variance (Figure 2a), respectively, while archaeal axes 1 and 2 explained 44.6% and 33.7% of the variance (Figure 2b), respectively. Fungal axes 1 and 2 explained 29.9% and 16.4% of the variance (Figure 2c), respectively. The permutational multivariate analysis of variance (PerMANOVA), function Adonis (vegan package for R), confirmed that water addition had significant effects on bacterial taxonomic communities (R^2^ = 0.422, *p* = 0.001, Appendix A) and archaeal taxonomic communities (R^2^ = 0.558, *p* = 0.001, Appendix A), and had weakly statistically significant effects on fungal taxonomic communities (R^2^ = 0.276, *p* = 0.057, Appendix A).

### 3.3. Responses of Microbial Taxa to Water Addition

As shown in Figure 3a, bacterial communities were dominated by Actinobacteria (41.6%), Proteobacteria (33.3%), Firmicutes (5.9%), Acidobacteria (3.4%), Chloroflexi (4.0%), Gemmatimonadetes (3.7%), and Bacteroidetes (2.5%). The relative abundances of these seven phyla accounted for more than 90% of the total relative abundance.

Water addition significantly increased the relative abundance of Proteobacteria from 29.7 ± 2.5% (control) to 39.5 ± 2.0% (W100) and that of Acidobacteria from 2 ± 0.1% (control) to 5.1 ± 0.4% (W100) (Appendix A). Intriguingly, we also noticed some functional, but rare taxa with significant increases in abundance (Appendix A). For example, the relative abundance of Planctomycetes, which contain a group of anaerobic ammonium-oxidizing (anammox) bacteria, increased significantly by 50.8–170.1% in water addition plots. The relative abundance of Nitrospirae, which contains a group of nitrite-oxidizing bacteria, increased considerably by 21.6–94.6% in water addition plots (Appendix A). On the contrary, the relative abundances of Actinobacteria significantly decreased from 43.9 ± 2.1% (control) to 34.7 ± 1.9% (W100). Firmicutes significantly decreased from 8.5 ± 1.5% (control) to 4.2 ± 0.3% (W100) and Bacteroidetes from 3.2 ± 0.3% (control) to 2.1 ± 0.1% (W100) (Appendix A).

To further investigate the taxonomic compositions of different soil bacterial communities, we compared the water addition response pattern of dominant bacterial taxa at the finer classification levels. At the genus level, a total of 581 genera were identified. However, only two genera significantly decreased in terms of abundance (Appendix A). One was the actinobacterial genus *Microbacterium* (*p* = 0.003), which can digest complex organic plant compounds and produce exopolysaccharides [45,46] while the other was from the proteobacterial genus *Rubellimicrobium* (*p* = 0.036), which is an extremotolerant bacterial taxon that disappears with improved environmental conditions [47]. No genus demonstrated significant increases as a response to water addition.

For archaeal communities, more than 99.7% of sequences could be classified into two phyla (Figure 3b), Thaumarchaeota (86.6%) and Euryarchaeota (13.1%), with the remaining sequences being classified into the newly found phyla, Woesearchaeota (0.2%, formerly Euryarchaeota DHVEG-6 cluster) and Bathyarchaeota (0.1%, MCG), as well as other minor phyla. Water addition had remarkable impacts on two dominant archaeal taxa, with Thaumarchaeota abundance increasing from 81 ± 1.7% (control) to 93.3 ± 1.2% (W100) and Euryarchaeota decreasing from 18.9 ± 1.7% (control) to 6.3 ± 1.0% (W100) (Appendix A). The general patterns of Thaumarchaeota were primarily driven by its major order Candidatus Nitrososphaera, which is associated with ammonia-oxidizing processes and increased from 15.3 ± 3.3% in control plots to 21.9 ± 3.3% in W100 plots. In contrast, the changes in Euryarchaeota were driven by the abundance of its dominant order Thermoplasmatales, which accounted for 99.9% in the division of Euryarchaeota (Appendix A). The relative abundance of Thermoplasmatales significantly decreased from 18.8 ± 1.7% in control plots to 6.3 ± 3.3% in W100 plots.

Ascomycota (83.8%) was the overwhelmingly dominant fungal group, followed by Basidiomycota (0.7%), Zygomycota (0.5%), and Chytridiomycota (0.3%) (Figure 3c)., About 14.6% of the sequences were unclassified. Furthermore, the ANOVA results showed that water addition had no effects on the relative abundances of any dominant fungal taxa at different class levels (e.g., phylum, order, class, and genus).

### 3.4. Relationships between Microbial Communities and Plant/Soil Properties

To link the microbial community structure with soil and plant properties, Mantel tests were applied to test the correlations between every environmental variable and bacterial, archaeal, and fungal communities (Figure 4). Among all the environmental variables examined, soil moisture at the depth of 20 cm (Figure 4) showed significant correlations with the soil bacterial community composition. However, no plant properties measured in our study were significantly correlated with soil bacterial community composition. Mantel tests were also applied to examine the potential correlations between individual bacterial phyla and environmental variables. For example, the compositions of Proteobacteria and Acidobacteria were both significantly correlated with soil moisture at the depth of 20 cm (Appendix A), while the composition of Gemmatimonadetes significantly correlated with soil temperature at the depth of 20 cm. In contrast to the bacterial community, the archaeal community composition showed a significant correlation with soil temperature at the depth of 20 cm (Figure 4) and a marginally significant correlation with soil NO_3_^−^. The composition of the most abundant phylum Thaumarchaeota strongly correlated with soil temperature (R = 0.309, *p* = 0.017; Appendix A). Another dominant phylum, Euryarchaeota had strong correlations with total nitrogen (R = 0.606, *p* = 0.001; Appendix A), total carbon (R = 0.669, *p* = 0.003), soil moisture (R = 0.732, *p* = 0.001), and plant coverage (R = 0.230, *p* = 0.019). The fungal community composition only showed a significant correlation with soil moisture at the depth of 20 cm (Figure 4), while the dominant phylum Ascomycota also correlated with soil moisture at the depth of 20 cm (R = 0.246, *p* = 0.051; Appendix A).

We further developed the structure equation model (SEM) to explore the association of the microbial community composition with multiple factors at different water addition levels (Figure 5). Parameters in the model included soil moisture, soil temperature, total nitrogen, total carbon, plant species, and plant coverage. Due to the collinearity of precipitation and soil moisture, precipitation was removed from model parameters. The primary ordination axes (PCoA1) of microbial communities were used to represent the microbial community compositions. The model for bacteria explained 46.5% of the variation in the bacterial community along the water addition gradient (Figure 5). Soil moisture and plant species richness were identified as the significant factors that shape the bacterial community structure. Standardized total effects (including direct and indirect effects) obtained from standardized SEM, suggested that soil moisture contributed most significantly to the variations in the bacterial community structure. The model for archaea explained 36.1% of the variation in the archaeal community along water addition gradients (Figure 5). The model showed that soil pH and NO_3_^−^ had significant direct adverse effects on the archaeal community structure (Figure 5). Similar to results of the Mantel test, soil moisture had no significant direct or indirect effects on the archaeal community composition. In addition, the model for fungi showed that water addition did not directly or indirectly affect the measured parameters of the fungal community composition, which was consistent with the results of the Mantel tests (Appendix A).

## 4. Discussion

Dry climates, scarce surface water, and strong evaporation characterize desert ecosystems, making them fragile and particularly sensitive to climatic changes [17]. In the present study, we used a long-term in situ manipulation model to investigate the effect of increased precipitation on soil bacterial, archaeal, and fungal communities in a desert ecosystem. The nutrient levels in the studied site were relatively low. Some soil variables, such as total nitrogen and total carbon, increased along the water addition gradient. This increase could be because the stimulation of plant growth under water addition increased the production of root exudates and litter input [31,48], which were important sources for soil carbon and nitrogen. In addition, this increase overall explained the increase in microbial biomass with increased precipitation. Furthermore, the compositions of bacterial and archaeal communities were both significantly altered by water addition, indicating that gradual (25%) increases in precipitation can substantially alter the soil microbial community composition (see also Bell et al., 2013, Clark et al., 2009, Austin et al., 2004 [13,26,49]). In particular, we observed a generalized increase in bacterial species richness with water addition, which was also recognized in previous desert studies [20,50]. This is most likely because some dormant bacterial taxa are revived when water stress and nutrient stresses are alleviated [51]. However, the Shannon diversity index and Pielou’s evenness index in this study experienced minimal change with water addition, most likely due to the uniform contribution of bacterial taxa to the overall abundance. On the contrary, within the archaeal community water addition decreased the Shannon diversity and Pielou’s evenness indices. This was attributed to the stimulation of several archaeal taxa by water addition (e.g., Candidatus Nitrososphaera of Thaumarchaeota, Figure 3b and Appendix A); whereby the proliferation of these archaeal taxa leads to a decrease in evenness at the community level.

Bacterial communities were dominated by Actinobacteria, Proteobacteria, Firmicutes, Acidobacteria, Chloroflexi, Gemmatimonadetes, and Bacteroidetes, accounting for more than 90% of the sequences in each of the examined soil samples. The dominant taxa were similar to those observed in other desert systems [6,9,52]. The dominance of Actinobacteria and Firmicutes can be explained by its endospore-forming ability and the high G+C content, enabling them to survive in challenging dry conditions [53,54]. Moreover, the phylum of Actinobacteria contains a number of OTUs capable of efficiently hydrolyzing complex substrates, such as starch, cellulose, pectin, and xylan [45], helping them to cope with nutrient-limited conditions. Another dominant phylum, Proteobacteria, had a large proportion of genera that are capable of nitrogen fixation and growing at low carbon or nitrogen concentrations [55]. Hence, in this study the dominant taxonomic groups were specialized in local nutrient-limited and water-limited environments (see also Shade et al., 2012 [56]).

Furthermore, the relative abundance of some specific taxa along the water addition gradient also changed significantly. For example, Proteobacteria and Acidobacteria were more abundant at higher water levels, while Actinobacteria was less abundant in the same plots. Our results are similar to those of previous studies of desert microbial communities [20,57,58], suggesting that water addition facilitates the growth of oligotrophic taxa and suppresses copiotrophic taxa in desert ecosystems. However, previous studies have suggested that increased water availability could stimulate the mineralization of soil organic matter [49,59] and promote the growth of copiotrophic microbes by providing more nutrients. This discrepancy could be attributed to two possible reasons. First, the nutrients released by mineralizing soil organic matter were limited due to low soil organic matter in the desert, therefore not supporting the copiotrophic taxa [20]. This explanation is supported by the low inorganic nitrogen (NO_3_^−^ and NH_4_^+^) concentrations in all plots. The contents of NH_4_^+^ were below the detection limit and NO_3_^−^ was about a twentieth of that in grassland [60,61]. As a result, water addition still favored oligotrophic microbes specializing in the use of recalcitrant organics over copiotrophic microbes. Moreover, as shown in previous studies, when the aridity index (AI, the ratio of precipitation to evapotranspiration) is less than 0.2, the relative abundances of oligotrophic taxa, such as Acidobacteria increased with an increasing AI value [20,58], suggesting that oligotrophic taxa are more sensitive to water addition in arid areas. The AI in our study site was 0.06, far below 0.2, and therefore, it is reasonable to see that the relative abundance of oligotrophic taxa increased while the copiotrophic taxa decreased with water addition.

Originally, nitrification was thought to include two main steps: the initial oxidation of ammonia to nitrite, and subsequently the oxidation of nitrite to nitrate [62]. Intriguingly, we observed that the relative abundance of the Planctomycetes phylum, which includes members capable of anaerobic ammonium oxidation (anammox) [63], increased significantly with water addition. Furthermore, Nitrospirae, which includes members known to convert nitrite to nitrate in nutrient-low sites [64], increased its abundance with water addition. The increased relative abundances of the above two phyla indicate that water addition facilitated nitrification, which can explain why NH_4_^+^ concentrations were below the detection limit, while NO_3_^−^ was stable across all treatments. Another reason for undetectable NH_4_^+^ is the plant uptake and the enhanced ammonia-volatilization due to a mild alkaline pH in the desert. Another aspect of NO_3_^−^ obtained from nitrification, is its consumption by plants for growth, this was supported by the increased plant cover that was observed across all experiments.

Water addition significantly increased the relative abundance of Thaumarchaeota but decreased Euryarchaeota in archaeal communities. Thaumarchaeota have recently received considerable attention as some members are known to perform ammonia oxidation under extremely low ammonium concentrations [65], which is also found in other desert soils worldwide [27,66,67]. For example, a recent study conducted in the Tarim Basin by analyzing microbial metagenomic and metatranscriptomic data suggested that Thaumarchaeota are the most important contributors to ammonia oxidation in desert soils [67]. Hence, in our study, the dominance of Thaumarchaeota suggests that the archaeal community in this desert was also dominated by ammonia-oxidizing archaea (AOA). At a closer level, the dominant group *Candidatus Nitrososphaera* within Thaumarchaeota [68] increased its relative abundance with water addition. This observation was attributed to the increased plant growth, which could provide higher organic substrates for *Candidatus Nitrososphaera* [69,70]. This matched a previous study that found *Candidatus Nitrososphaera* performed better in organic-rich habitats [71]. In contrast to the increase in Thaumarchaeota, the relative abundance of Euryarchaeota decreased by water addition (13.1% in the control to 6.9% in W100). Interestingly, all the detected euryarchaeal OTUs were classified as Thermoplasmatales at the order level. However, the order of Thermoplasmatales in desert ecosystems remains poorly characterized [72,73], which calls for further investigations.

Our results show that fungal alpha-diversity and community composition were only weakly affected by water addition, this was similar to previous findings in grassland sites, where the relative abundance of various fungal taxa were relatively robust to rewetting [15]. As suggested by previous physiological studies, fungi can produce hyphae at low volumetric soil moisture contents [74], which can help them access water and nutrients from distant micropores under dry conditions. Thus, fungal taxa exhibit a high tolerance to water stresses and changes in desert soils [75,76].

In the present study, we also used Mantel tests to establish linkages between bacterial, archaeal, and fungal communities and environmental variables; we also performed SEM to provide a mechanistic basis for our understanding of how environmental variables mediate alterations in soil microbial community compositions. The bacterial and archaeal communities were both affected by environmental variables; the shifts in the bacterial community composition were primarily caused by soil moisture changes. Similarly, the SEM results suggest that moisture, as well as plant species, directly influenced bacterial communities. Both of the above results indicate that soil moisture was the main factor regulating bacterial communities. This result is in line with a previous study that investigated the distribution of bacteria along a precipitation gradient; it was shown that a change in soil moisture was the main factor that shaped the microbial community composition in a desert ecosystem [77].

The Mantel test showed that shifts in archaeal community composition were mainly driven by changes in temperature and NO_3_^−^ concentrations. Similarly, based on the SEM results, the NO_3_^−^ concentrations directly affected archaeal communities. Both the above results suggest that the NO_3_^−^ plays a critical role in shaping archaeal community compositions, although the results from different tests (e.g., Mantel test and SEM) differed slightly. It has also been stated that nitrogen is the second most important factor for desert ecosystems [66]. Our results were similar to those of a study conducted in semi-arid regions in Inner Mongolia, which showed that archaeal community changes were closely correlated to nitrogen contents [66], most likely because major taxonomic groups such as Thaumarchaeota (AOA) are sensitive to soil nitrogen content [24,78]. Although we included all measured environmental variables in the model, we could not find the pathway by which water addition affected archaeal communities. One possible reason is that archaeal community and environmental variables changed nonlinearly along the water addition gradient while another possibility is that some deterministic environmental variables affecting microbial communities in the desert ecosystem were not measured in this study, such as soil electrical conductivity and other soil physico-chemical parameters [66,79,80].

Archaeal and bacterial communities responded in different ways to water addition, probably due to their distinctive niches in desert environments. It has been suggested that soil niches play essential roles in structuring communities [66,81]. Archaea and bacteria occupy different niches, and niches mediating bacterial responses were independent from those mediating archaeal responses [21,66,77]. This is consistent with our findings that bacteria were mainly affected by soil moisture, whereas the archaeal community was mainly affected by soil nitrogen contents. Thus, the distinct environmental preferences of bacteria and archaea resulted in different responses of these microbial groups under precipitation changes.

## 5. Conclusions

We provide novel information on long-term water addition effects on soil bacterial, archaeal, and fungal communities in a desert ecosystem. Water addition induced significant changes in bacterial community composition, with increasing abundance of oligotrophic taxa and decreasing abundance of copiotrophic taxa. Water addition also significantly altered archaeal community compositions, increasing the relative abundance of Thaumarchaeota and decreasing that of Euryarchaeota. However, water addition did not alter fungal communities. Our results further suggest that soil moisture is the main factor that drives the shift of bacterial communities, while the soil nitrate content is the main factor driving the archaeal community composition. Given the predominant role of microbes in nutrient cycles, our findings deepen our understanding of the imbalance between nutrient cycles induced by increased precipitation, helping us to assess the systematic ecological responses to climate change.

## Figures and Tables

**Figure 1 microorganisms-09-00981-f001:**
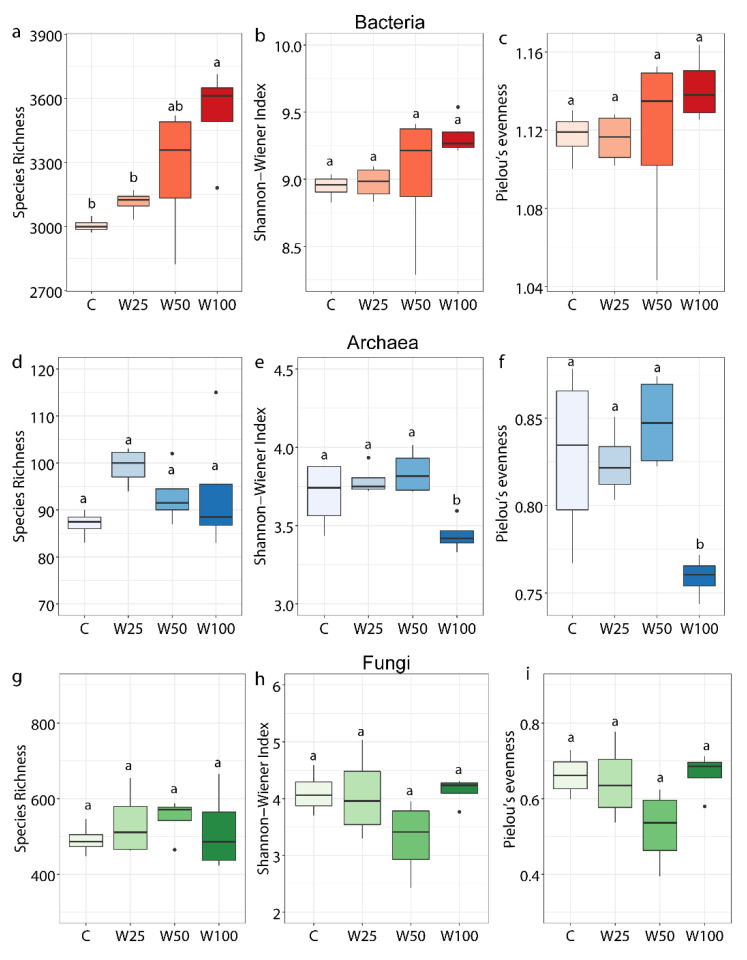
Alpha-diversity of bacterial (**a**–**c**), archaeal (**d**–**f**), and fungal (**g**–**i**) taxonomic communities in different water addition treatments. Boxes represent values from the lower 1/4 quantile to the upper 1/4 quantile. The tops and bottoms of boxes represent the upper 95% CI and the lower 95% CI, respectively. Outliers are drawn as black solid circles above or below boxes. Lowercase letters depict significant differences across treatments. C = ambient precipitation; W25 = ambient precipitation +25% of local annual mean precipitation; W50 = ambient precipitation + 50% of local annual mean precipitation; W100 = ambient precipitation +100% of local annual mean precipitation.

**Figure 2 microorganisms-09-00981-f002:**
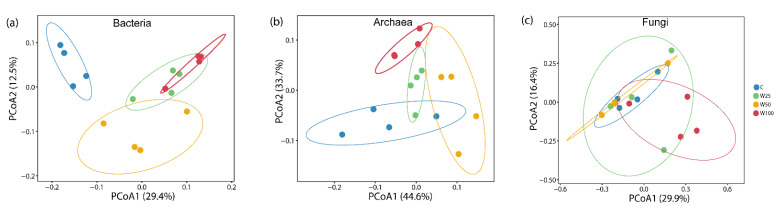
PCoA plots of Bray–Curtis dissimilarities highlighting that bacterial (**a**), archaeal (**b**), and fungal (**c**) communities were significantly different in composition depending on water addition treatments. Circles represent samples from each treatment. C = ambient precipitation; W25 = ambient precipitation +25% of local annual mean precipitation; W50 = ambient precipitation +50% of local annual mean precipitation; W100 = ambient precipitation +100% of local annual mean precipitation.

**Figure 3 microorganisms-09-00981-f003:**
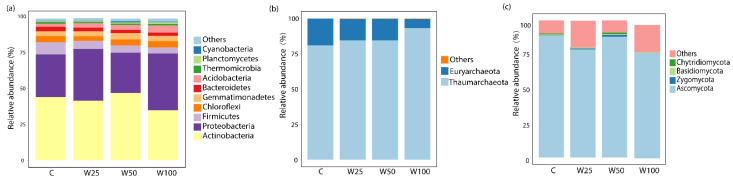
Relative abundances of microbial phyla in bacterial (**a**), archaeal (**b**), and fungal (**c**) communities. C = ambient precipitation; W25 = ambient precipitation +25% of local annual mean precipitation; W50 = ambient precipitation +50% of local annual mean precipitation; W100 = ambient precipitation +100% of local annual mean precipitation.

**Figure 4 microorganisms-09-00981-f004:**
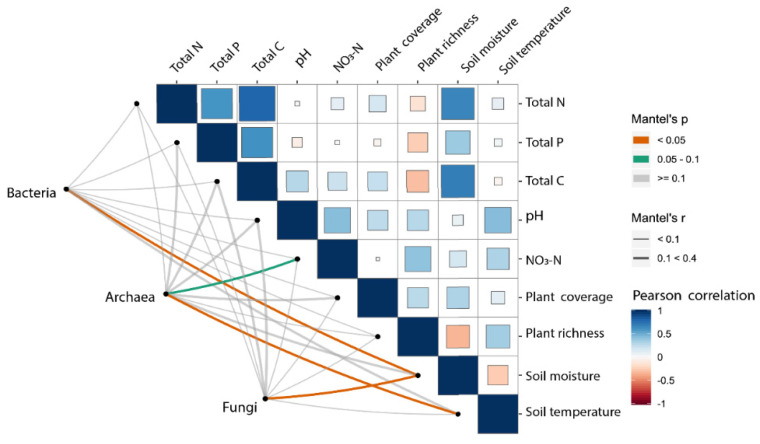
Mantel’s correlations between the environmental factors and bacterial, archaeal, and fungal taxonomic communities in different water addition treatments (The widths of lines represent the magnitudes of Mantel’s r statistics and the colors indicate the significance of test; the sizes of colored blocks denote the significance of Pearson correlation between two environmental factors, and the absolute values of the Pearson correlation coefficient were labelled along the colored bar on the right).

**Figure 5 microorganisms-09-00981-f005:**
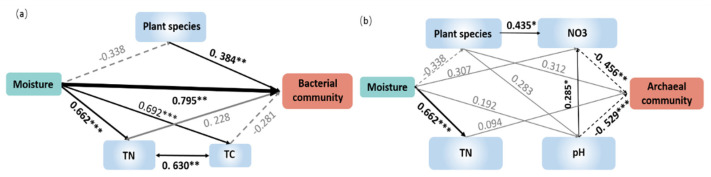
Structural equation modeling showing the relationships between plant/soil properties and the bacterial (**a**) and archaeal (**b**) community compositions. Solid arrows indicate positive effects, and the dashed arrow indicates a negative correlation. The standardized path coefficients are adjacent to the arrows and indicate the effect size of the relationship. Arrow widths are proportional to the strength of each relationship. Percentages beside the response variables refer to the proportion of variance explained by the model (R^2^). Results of model fitting: (**a**) bacteria: χ^2^ = 0.449, df = 2, *p* = 0.799; CFI = 1.000; AIC = 36.449; RMSEA = 0.000, *p* = 0.806; (**b**) archaea: χ^2^ = 1.833, df = 4, *p* = 0.766; CFI = 1.000; AIC = 35.834; RMSEA = 0.000, *p* = 0.779. TN, soil total nitrogen; TC, soil total carbon; Temperature, soil temperature at the depth of 20 cm; Moisture, soil moisture at the depth of 20 cm. * *p* < 0.05, ** *p* < 0.01, and *** *p* < 0.001.

**Table 1 microorganisms-09-00981-t001:** Effects of water addition on soil and aboveground plant properties. Values represent mean values and standard deviations (*n* = 4).

	C	W25	W50	W100
September mean soil moisture (20 cm, %)	3.17	3.20	3.85	6.79
September mean soil temperature (20 cm, °C)	23.58	24.28	20.09	22.36
NO_3_^−^ (×10^−3^ g·kg^−1^)	**1.50 ± 0.28 a** ^1^	**1.19 ± 0.23 a**	**0.86 ± 0.06 a**	**1.49 ± 0.38 a**
TC ^2^ (g·kg^−1^)	**0.10 ± 0.02 b**	**0.10 ± 0.01 b**	**0.10 ± 0.01 b**	**0.16 ± 0.05 a**
TN (g·kg^−1^)	**0.14 ± 0.02 b**	**0.15 ± 0.01 b**	**0.14 ± 0.01 b**	**0.18 ± 0.03 a**
TP (×10^−4^ g·kg^−1^)	1.37 ± 0.05 a	1.43 ± 0.05 a	1.39 ± 0.09 a	1.51 ± 0.07 a
pH	7.89 ± 0.031 a	7.89 ± 0.045 a	7.85 ± 0.03 a	7.89 ± 0.02 a
Plant coverage (%)	25.0 ± 7.07 a	31.25 ± 4.79 a	26.25 ± 11.09 a	33.75 ± 9.47 a
Plant species	7.25 ± 1.50 a	8.25 ± 0.96 a	7.00 ± 0.82 a	6.75 ± 0.96 a

^1^ Values in bold indicate significant responses to water addition, as assessed by one-way ANOVA. Lowercase letters in bold depict significant differences across treatments. ^2^ Abbreviations: TC, total carbon; TN, total nitrogen; TP, total phosphorus; C = ambient precipitation; W25 = ambient precipitation +25% of local annual mean precipitation; W50 = ambient precipitation +50% of local annual mean precipitation; W100 = ambient precipitation +100% of local annual mean precipitation.

## Data Availability

Sequence reads generated in this study were archived in the sequence read archive database of the National Center for Biotechnology Information under accession number PRJNA563974.

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
