# Peer review of "The Responses to Long-Term Water Addition of Soil Bacterial, Archaeal, and Fungal Communities in a Desert Ecosystem"

_microorganisms, 2021, doi:10.3390/microorganisms9050981_

Round 1

Reviewer 1 Report

I do find this work interesting and valuable. The manuscript is really well written. I have only few suggestions (mostly editorial) to improve the manuscript:

Introduction

Line 63 – “fungal” should start with capital letter.

Material and methods

Line 139 – “comers” or corners?

Results:

Table 3 should be Archaea not “Achaea”

Discussion

Line 413 – lack of a dot in the end of the sentence

Reviewer 2 Report

MDPI Microorganisms (Microorganisms-1192912): "The responses to long-term water addition of soil bacterial, archaeal and fungal communities in a desert ecosystem".

The subject of the manuscript is consistent with the scope of the Journal. Manuscript present many interesting results about an important subject. The present paper is prepared in the usual manner for scientific work, both the division into chapters collected results in the form of tables and figures. The authors applied correct analytical methods and received many interesting results. The obtained results do not raise any substantive objections. In my opinion, that the work is suitable for printing journal Microorganisms. Minor notes below:

  1. Not all abbreviations are explained under figures.
  2. The some figures are of poor quality and not very clear (it must be corrected).
  3. Please, be sure that all the references cited in the manuscript are also included in the reference list and vice versa with matching spellings and dates.

Reviewer 3 Report

Excellent work.
I have only few questions to the authors.
Considering the soil sampling 5 individual samples were pooled in each plot. The root densities may be different, which have significant effect on microbial biomass and diversity. How can you consider the heterogeneity originated from inequal distribution of roots?

In Table S1 the Permanova pairwise comparison is shown. As I know the Adonis (permanova) in R can not compute pairwise comparison. Are you applied pairwise.adonis package or other function? 

In discussion part
Two possible explanation was given for undetectable ammonium, one is the restricted mineralization of organics the other is the increased nitrification rate. A third would be in my opinion the most possible, the plant uptake and the enhanced ammonia-volatilization due to a mild alkaline pH?

In conclusion part "the soil nitrite content is the main factor driving the archaeal community composition" was written, however soil nitrite was not measured, but  nitrate. 

Others:
L139 "corners instead of comers" 
